# The Effect of Fatty Acids Profile in Potato and Corn Chips on Consumer Preferences

**DOI:** 10.3390/foods13203292

**Published:** 2024-10-17

**Authors:** Okan Gaytancıoğlu, Fuat Yılmaz, Ümit Geçgel

**Affiliations:** 1Department of Agricultural Economics, Faculty of Agriculture, Tekirdağ Namık Kemal University, 59030 Tekirdağ, Türkiye; fuatyilmaz@nku.edu.tr; 2Department of Food Engineering, Faculty of Agriculture, Tekirdağ Namık Kemal University, 59030 Tekirdağ, Türkiye; ugecgel@nku.edu.tr

**Keywords:** chips, fatty acid composition, consumer behavior, factor analysis

## Abstract

The global market for potato and corn chips is rapidly expanding due to the modern fast-paced lifestyle. However, the high fat content, especially saturated fats in these deep-fried snacks, poses significant health risks such as hypertension, coronary heart disease, and diabetes. In this study, fatty acid profiles of commercially available corn and potato chips are analyzed and their impacts on consumer preferences in Turkey is examined. The findings reveal notable differences in the nutritional content between potato and corn chips, with potato chips generally having higher fat and protein content. The survey results indicate that consumer preferences are significantly influenced by age, education level, and occupation. The factor analysis identified three main components affecting purchasing decisions: nutritional value and additives, hygiene and brand quality, and price and affordability. Considering these insights, manufacturers should be encouraged to reformulate their products to meet the increasing demand for healthier options, emphasize food safety standards, and balance product quality with affordability to appeal to a broader range of consumers.

## 1. Introduction

The potato and corn chips market holds significant importance both in Turkey and globally, appealing to a wide consumer base. Since 2020, the snack food industry has experienced substantial growth, propelled by modern, fast-paced lifestyles that make convenience foods more appealing. Chips have emerged as one of the preferred snacks worldwide [1]. Potato and corn chips are frequently consumed as part of our everyday diet [2]. However, most crunchy snack chips are produced through deep-frying in hot oil, resulting in high fat content products. Excessive consumption of these fats, especially saturated fats, increases the risks of various health problems such as coronary heart disease, diabetes, cancer, and hypertension [3]. These unhealthy factors were primarily attributed to the high fat and sodium content versus low fiber, vitamin, mineral, and protein composition of snack food [4].

Increasing health awareness among consumers is significantly influencing their purchasing behavior, steering them toward healthier food options. Studies from countries like Thailand, India, Italy, and Bangladesh highlight this shift in consumer behavior [5,6,7,8,9]. The food industry is continually evolving to meet these changing preferences and demands. With growing awareness of health and fitness, there is a rising demand for snacks that not only satisfy taste but also offer nutritional benefits. This trend has led manufacturers to develop products with improved fatty acid profiles, reduced saturated fats, and enhanced nutritional composition [10].

Previous research has examined the impacts of fatty acid compositions and the nutritional quality of edible snacks, particularly potato and corn chips, on consumer preferences from different perspectives. For instance, a study in Romania analyzed the fatty acid profile of French fries and potato chips, concluding that these products have low trans fatty acid concentrations, making them suitable for human consumption [11].

Fried foods, especially potato and corn chips, are popular worldwide. However, the deep-frying process can lead to the formation of harmful trans fatty acids (TFAs) due to oil degradation. Studies from various regions highlight the presence of TFAs in fried snacks, such as plantain chips in Ghana and potato chips in India, both of which showed significant TFA formation during frying cycles [12,13]. The health risks associated with TFAs, including increased risks of coronary heart disease and diabetes, are well documented [12,14]. Therefore, understanding the fatty acid profiles of chips and identifying healthier frying practices is crucial for consumer safety.

In Brazil, a study investigated consumer attitudes toward various potato varieties and found that while consumers had insufficient knowledge about different varieties, their attitudes toward price and familiarity were significant determinants in purchasing decisions [15]. Similarly, research from the USA examined the effects of nutritional profiles on consumer demand for chips, concluding that sodium and saturated fat enhanced consumer satisfaction, whereas high energy and total fat reduced satisfaction [16].

Additionally, studies have explored consumer preferences for classic potato chips compared to organic vegetable chips. Findings indicate that socio-demographic factors such as consumption frequency and education level significantly impact preferences, suggesting that manufacturers are adapting their products to meet these evolving demands [17].

Despite these insights, there is still a need to further investigate the fatty acid composition of potato and corn chips and their impact on consumer preferences, particularly in different markets. Understanding these factors is crucial for developing healthier snack options and guiding consumers toward better nutritional choices.

This study aims to identify and compare the fatty acid composition of corn and potato chips and to evaluate their impacts on consumer preferences. By supplying a detailed analysis of the nutritional profiles of these popular snacks and assessing consumer attitudes, this research seeks to contribute to the development of healthier snack alternatives and inform both manufacturers and consumers about the importance of nutritional quality in snack foods.

## 2. Materials and Methods

### 2.1. Materials

The most common five brands of corn and five brands of potato chips were purchased from Istanbul’s local market for content and composition analysis. They were kept in the refrigerator under dry and dark conditions until the test.

To evaluate consumers’ purchasing preferences, an online survey was applied to individuals living throughout Turkey and covering different age groups. Participants were invited via social media and email lists. Participation was voluntary and randomly selected without any gender, age, or education level restrictions. The sample size is calculated as 667 using Equation (1) [18].
(1)$$\begin{array}{c} n=\frac{{p*q*z}_{\propto /2}^{2}}{D^{2}}\\ n=\frac{0.5*0.5*2.58^{2}}{0.05^{2}}≅\;667 \end{array}$$
where

n represents the sample size,

p denotes the probability that a unit is in the population that consume chips (0.5),

q equals 1 − p, representing the proportion of those not consuming chips (0.5),

z_(∝/2)_ is the confidence coefficient (for a 99% confidence interval, this value is 2.58),

D refers to the margin of error, set at 5%.

To achieve the highest sample size, p and q are assumed to be 0.5.

### 2.2. Laboratory Analysis

#### 2.2.1. Fat Content

Crude fat analysis of corn and potato chips samples was carried out according to the Soxhlet extraction principle and using hexane solvent [19] methods no. 1.121.

#### 2.2.2. Protein Content

Seed samples underwent protein analysis through the Kjeldahl method, a technique for nitrogen determination prescribed by the International Association of Official Analytical Chemists [20]. The conversion factor of nitrogen-to-protein content utilized is calculated as 6.25.

#### 2.2.3. Ash Content

The method for determining ash content complied with the specifications of the International Organization for Standardization [21] under method 749. Dried seed samples were subjected to ignition and subsequent incineration using a muffle furnace in which temperatures were systematically increased to 550 °C. The process continued until a stable mass was reached for the seed samples.

#### 2.2.4. Preparation of FAME

Utilizing the Ce 2-66 method outlined by the American Oil Chemists’ Society [22], Fatty Acid Methyl Esters (FAMEs) were synthesized from the oils of potato and corn chips through alkaline hydrolysis, followed by a methylation process using a 12.5% solution of BF3 in methanol. The resulting FAMEs had a concentration of approximately 7 mg/mL in heptane. High-purity FAME standards, at 99%, were sourced from Nu-Chek-Prep Inc. in Elysian, MN. The compositional analysis was performed with a capillary gas-liquid chromatography (GLC) setup using a HP 6890 GC (Hewlett-Packard, CA, USA) system that includes a split injector and a flame ionization detector (FID). For this analysis, a Chrompack capillary column, CPTM-Sil 88, with a length of 100 m, an internal diameter of 0.25 mm, and a film thickness of 0.2 µm, was used. The GLC’s temperature program began at 130 °C, maintaining this temperature for five minutes before increasing at a rate of 2 °C per minute up to 177 °C. Both the injector and detector were maintained at temperatures of 225 °C and 250 °C, respectively. Helium served as the carrier gas, flowing at a rate of 1 mL per minute.

### 2.3. Statistical Analyses

All characteristics examined in the experiment, which was set up according to five corn and potato chips, were subjected to variance analysis according to the “Random Plot Trial Design”. Difference groupings of their means are presented in separate tables. All characteristics in which corn and potato samples showed significant differences were subjected to the LSD test at the 0.05 significance level.

Crosstabs and Chi-square tests were used to examine the relationships between demographic variables (gender, age group, education level, occupation) and chips preferences, consumption frequency, and factors influencing purchasing decisions. Chi-square tests were utilized to assess the statistical significance of the observed relationships.

Factor analysis was performed to determine the main factors that shape the correlation patterns among variables affecting chip purchasing decisions. To extract and interpret these factors, Principal Component Analysis (PCA) with a Varimax rotation was employed. The suitability of the data for factor analysis was assessed using the Kaiser–Meyer–Olkin (KMO) measure.

## 3. Results

### 3.1. Fatty Acid Composition of Corn and Potato Chips

According to Table 1 and Table 2, as well as Figure 1 and Figure 2 the fat content in corn chips ranged from 19.89% to 28.92%, while in potato chips, it ranged from 28.57% to 34.58%. The protein content in corn chips was found to be between 4.90% and 5.26%, whereas in potato chips, it ranged from 5.81% to 6.82%. The ash content for corn chips varied from 0.96% to 3.16%, compared to 4.35% to 5.57% in potato chips.

The saturated fatty acids (SFAs) in corn chips ranged from 7.75% to 45.54%, and in potato chips, from 12.84% to 45.74%. The monounsaturated fatty acids (MUFA) content in corn chips was between 43.62% and 65.94%, while in potato chips, it ranged from 43.66% to 52.15%. The polyunsaturated fatty acids (PUFAs) in corn chips ranged from 10.19% to 26.30%, and in potato chips, from 10.10% to 37.66%.

These findings highlight the differences in nutritional content between corn and potato chips, emphasizing the higher fat, protein, and mineral content in potato chips compared to corn chips. Both types of chips contain significant amounts of monounsaturated, saturated, and polyunsaturated fatty acids.

This study analyzed the fatty acid composition, fat content, protein content, and ash content of industrially produced corn and potato chips. These results indicate that potato chips generally contain higher levels of fat compared to corn chips, contributing to the overall caloric density of these snacks and raising concerns regarding their role in diet-related health issues such as obesity and cardiovascular diseases.

While both types of chips offer minimal protein, this nutrient is crucial for muscle maintenance and overall bodily functions. The ash content, representing the total mineral content, was also higher in potato chips, suggesting they may provide more minerals (such as calcium, potassium, phosphorus, magnesium, etc.) though the overall health benefits should be considered in the context of their high fat content [1,2].

Both chip varieties contain significant amounts of monounsaturated fatty acids (MUFAs), saturated fatty acids (SFAs), and polyunsaturated fatty acids (PUFAs). High SFA intake increases the risk of heart disease, highlighting a potential health risk from frequent consumption of these snacks. MUFAs are known for their beneficial effects on cardiovascular health by lowering LDL cholesterol levels. However, the PUFA content, which includes ω-3 and ω-6 fatty acids, was also significant and can decrease heart disease risks when consumed in appropriate ratios with SFAs.

C18:2 may be protective against lacunar infarction and ischemic stroke, by lowering blood pressure and improving small vessel circulation, reducing platelet aggregation and increasing erythrocyte flexibility. Also, C18:2 is a highly potent anticancer fatty acid, showing a significant protective effect even in amounts as small as 1% of the diet [23].

Several factors, such as the composition of the frying oil, the type and makeup of the food, as well as frying conditions such as temperature and duration, can impact the heat and mass transfer between the food and the frying oil, ultimately affecting fat absorption and the TFA content [12].

As noted in Table 1 and Table 2, none of the potato and chip samples examined contained trans fatty acids. Given the widespread presence of trans fatty acids in numerous food products, their absence in the analyzed samples is notably important for public health. Trans fatty acids adversely impact total cholesterol levels to HDL cholesterol, having twice the detrimental effect compared to SFAs. They also raise LDL cholesterol levels while lowering HDL cholesterol, thus increasing risks of cardiovascular disease. Epidemiological research has consistently demonstrated a correlation between the consumption of trans fatty acids and an increased risk of cardiovascular issues. As a result, it is important to closely assess the potential health risks linked to excessive intake of trans fatty acids [24].

### 3.2. Survey Data Analysis

The demographic profile of the study sample indicates a diverse composition across age, gender, educational attainment, and occupational background (Table 3). The age distribution is predominantly skewed toward younger individuals, with 42.6% of partici-pants aged between 19 and 24 years, followed by 17.1% in the 25–34 age group, and 18.6% in the 35–50 range. Individuals aged 18 and under constitute 13% of the sample, while those aged 51 and above represent 8.7%. Gender distribution reveals a higher proportion of females (61.6%) compared to males (38.4%), suggesting a gender imbalance in the sam-ple. In terms of educational background, most participants were university students (33.6%), followed closely by university graduates (24.6%) and high school graduates (24.1%). Participants with middle school education comprised 7.6%, those with primary school education made up 4.8%, Master’s degree holders accounted for 4.2%, and only 1% have attained a PhD. The occupational structure was dominated by students, who represented 43.9% of the sample, followed by private sector employees (15.3%) and civil servants (7.6%). Freelancers accounted for 10.5%, while retired individuals constituted 4.2% of the sample. Homemakers, teachers, and academicians were less represented, with proportions of 3.6%, 0.6%, and 0.9%, respectively. An additional 13.3% of participants were engaged in various other professions.

Cross-tabulation analyses and Chi-square tests were used to examine the relationships between demographic factors and chip preferences. Analysis across different age brackets demonstrates distinct preferences in chips types and consumption frequencies. Individuals aged 19–24 show a pronounced preference for corn chips, while those aged 35–50 favor potato chips. The Pearson Chi-square test results (χ^2^ = 42.101, *p* < 0.001) confirm a statistically significant relationship between age group and type of chips preferred. Additionally, frequency of consumption varies notably with age; the 19–24 age group typically consumes chips 2–3 days per month, contrasting with those under 18 who consume chips approximately once a month (χ^2^ = 91.416, *p* < 0.001). Furthermore, flavor preferences also align with age differences; the 19–24 age group mostly prefers spicy chips, whereas those under 18 favor plain chips (χ^2^ = 318.535, *p* < 0.001).

Educational attainment further influences chips preferences. University graduates display a stronger preference for corn chips compared to primary school graduates who exhibit no marked preference between chip types (χ^2^ = 33.707, *p* = 0.014). The frequency of chips consumption also differs according to educational level; university students consume chips 2–3 days per month, whereas doctoral graduates tend to consume them less frequently, showcasing a statistically significant correlation (χ^2^ = 64.593, *p* = 0.002).

Occupational roles significantly affect both type and flavor preferences of chips. Students are inclined towards corn chips, while retirees show no strong preference between different types (χ^2^ = 288.822, *p* = 0.001). Regarding flavor preferences, students demonstrate a notable preference for spicy chips, in contrast to retirees who like them less (χ^2^ = 5901.841, *p* < 0.001).

Figure 3 shows the percentage distribution of consumer preferences based on various product attributes, including price, taste, packaging, calories, nutritional value, additives, origin, hygiene, brand, organic certification, and GMO-free labeling. The results indicate that certain factors, such as taste, hygiene, and brand, play a dominant role in influencing consumer choices. For instance, 64.2% of consumers “strongly agree” that taste is a crucial factor in their purchasing decisions, highlighting its significant impact on product preference. Similarly, hygiene appears to be a decisive attribute, with 69.6% of consumers “strongly agreeing” that it is important, followed by 58.9% who consider brand as a key factor. In contrast, attributes such as cheapness and calories show a more divided response, with a notable portion of consumers expressing “no opinion” (27.7% for cheapness, 21.9% for calories), suggesting that these factors may not be universally important across all segments. Additionally, the table reveals that factors like organic certification and GMO-free labeling are gaining traction, with 31.2% and 47.2% of consumers, respectively, “strongly agreeing” on their importance. This may reflect growing consumer awareness of health and sustainability issues. On the other hand, the attributes related to packaging and additives display more mixed reactions, implying that they are of moderate importance to consumers.

### 3.3. Factor Analysis

Before proceeding with the results of the factor analysis, we assessed the adequacy of the sample size and the presence of correlations among variables to determine if the data are suitable for factor analysis.

A sample size adequate for factor analysis is typically recommended to be four to five times the number of variables being studied. In this research, 667 participants responded to 12 different statements, demonstrating that the sample size is appropriate. To further assess the sample’s suitability, the Kaiser–Meyer–Olkin (KMO) test was applied. A KMO value between 0.5 and 1 is considered acceptable for such analyses. This study obtained a KMO value of 0.879, confirming that the sample size is sufficient for analysis, as indicated in Table 4 [25].

According to Table 5, three components explain 59.161% of the overall variance. The first component explains 39.320%, the second component explains 10.976%, and the third component explains 8.865% of the variance. Since three components explain a considerable portion of the total variance, a three-factor structure can be accepted. These results are supported by the scree plot embodied in Figure 4 where three components with eigenvalues bigger than 1 are selected.

Factor loadings indicate which factors the variables load more strongly on. Varimax rotation clarifies the factor loadings.

Factor 1, accounting for 39.320% of the total variance, emerged as the most effective factor. Initially, analysis without rotation revealed multiple variables loading on various factors. To clarify and facilitate this, a varimax rotation was applied, which reduces the prevalence of variables loading high on a single factor and increases the clarity of factor interpretations [25,26]. It is recommended to preserve factor loadings with absolute values exceeding 0.45 [25,27,28]. Table 6 shows the factor loadings after varimax rotation. Variables such as packaging, calories, nutritional value, additives, origin, organic certification, and GMO were grouped under Factor 1; Factor 2 included taste, hygiene, and brand; Factor 3 is primarily associated with price (Figure 5).

The factor analysis results show that there are three main factors that affect consumers’ preferences for chips: nutritional value and additives, hygiene and brand quality, and price and affordability.

The first factor, nutritional value and additives, reflects consumers’ increasing awareness of health-conscious choices. Consumers are increasingly paying attention to fatty acid profiles, the presence of additives, and certifications such as organic labels. This suggests that health-oriented consumers are more likely to choose chips that are perceived as healthier, have fewer artificial ingredients, and have a better nutritional composition. The increasing demand for products with enhanced nutritional value highlights a shift towards healthier lifestyles among a significant portion of consumers.

The second factor, hygiene and brand quality, emphasizes the importance of trust in food safety and brand reputation. While hygiene reflects consumers’ concerns about food safety standards, brand quality relates to their loyalty and trust in well-established brands. Well-known brands that emphasize high hygiene standards are likely to attract more consumers, as these features are closely linked to perceptions of product safety and reliability. This factor indicates that consumers prioritize chips from brands known for their quality assurance and food safety.

The third factor, price and affordability, addresses economic considerations in consumer purchasing behavior. For many consumers, especially those in lower income groups, price remains a critical determinant in their decision-making process. The affordability of chips can drive preferences toward more affordable options, making it a significant competitive factor among brands. However, there is also a segment of consumers who are willing to pay a premium for products with higher nutritional value or better brand reputation, suggesting a price–quality trade-off in purchasing decisions.

These three factors shape consumer behavior, with different levels of importance placed on each by different demographic groups. For example, younger consumers may prioritize taste and price, while older consumers may focus more on nutritional value and hygiene. This detailed understanding of consumer preferences provides manufacturers with valuable insights to develop targeted marketing strategies that meet the specific needs and values of their diverse customer base.

## 4. Conclusions

This study aimed to assess the nutritional quality and safety of potato and corn-based chips sold in the local Turkish market. The major fatty acids in potato and corn chips are detailed in Figure 1 and Figure 2. As demonstrated, corn chips exhibit varying levels of saturated fatty acids (SFAs), monounsaturated fatty acids (MUFAs), and polyunsaturated fatty acids (PUFAs). Potato chips, on the other hand, consistently show a higher proportion of PUFAs, which is indicative of their potentially healthier fatty acid profile when compared to corn chips. These differences in fatty acid content can influence consumer preferences, as indicated by the consumer survey responses analyzed in this study.

The factor analysis identified key components such as nutritional value, which includes fatty acid profiles, and tastes, which are influenced by the types of fats used. The preference for potato chips with higher PUFA content aligns with the growing consumer trend towards healthier dietary choices, reflecting an increased awareness of the health impacts of various types of fatty acids.

The factor analysis provided further insights by identifying three main factors that underpin consumer decisions when purchasing chips. These factors were labeled as follows: Nutritional Value and Additives, Hygiene and Brand Quality, and Price and Taste. The prominence of these factors underscores the importance of health considerations, perceived safety and reliability, and economic aspects in consumer decision-making.

The analysis of mineral content in potato and corn chips revealed a diverse composition of macro and microelements, contributing to the nutritional profile of the products. Potassium (K), phosphorus (P), and magnesium (Mg) were the most abundant macroelements in both chip varieties, with potassium levels reaching as high as 11,671 mg/kg in potato chips, and phosphorus levels of up to 1583 mg/kg in corn chips [1,2]. This aligns with previous studies, which highlighted potassium as a major component in fried potato products [2].

The data revealed that both corn and potato chips maintained their oxidative and hydrolytic stability over the shelf life, with minimal development of rancid flavors. However, the high levels of fat, particularly SFAs, coupled with the caloric density of these snacks, suggest that their consumption should be moderated. To balance their diets, consumers should pair these snacks with foods rich in ω-3 fatty acids and maintain an overall balanced diet rich in fruits, vegetables, and whole grains.

Given these findings, it is advisable to consume both corn and potato chips in moderation, considering their high fat and calorie content. Consumers should balance their diets with foods rich in ω-3 fatty acids, such as flaxseeds and fish, to mitigate the risks associated with high omega-6 and saturated fat intake from chips. Furthermore, opting for healthier snack alternatives, such as baked chips or vegetable crisps, can contribute to a more balanced diet. Regularly reading nutrition labels to monitor fat content and avoiding trans fats are also recommended practices.

In our research, none of the potato and chip samples examined contained trans fatty acids. However, in Addo et al. [12], the TFA concentrations varied across the samples, with mean levels ranging from 1.41 g/100 g of fat to 2.88 g/100 g of fat and 0.41 g/100 g of food to 0.78 g/100 g of food.

The fatty acid composition of potato chips changed significantly depending on the frying temperatures. In fact, Tajer and Özdemir showed in their study that sunflower oil led to higher saturated fat absorption compared to oleogels [29]. This is consistent with findings from studies using rice bran and beeswax oleogels, which showed a reduction in oil uptake and improved frying stability [14,29]. Additionally, the formation of starch-lipid complexes in chips fried in sunflower oil was less pronounced than in those fried in oleogels, similar to the results of studies on intermittent frying and the impact of unsaturated oils [30]. Further, Thakur et al. showed that chips fried in oleogels absorbed 23% less oil compared to those fried in conventional oils [14]. The use of rice bran wax as an oleogelator proved effective in reducing fat uptake and maintaining taste during frying, as previously demonstrated [31].

The analysis is limited to a sample of commercially available chips in Turkey, which may not fully represent global market variations. Additionally, the study focuses primarily on potato and corn chips, potentially excluding other snack categories that could offer further insights into fatty acid compositions.

In summary, while corn and potato chips can be enjoyed as part of a varied diet, attention to portion size and frequency of consumption is crucial. By integrating these snacks mindfully, alongside other nutrient-dense foods, individuals can enjoy them without compromising their overall health. These insights provide valuable guidance for both consumers and manufacturers in promoting healthier eating habits and developing better snack options.

Considering these results, manufacturers can implement several practical strategies to align with consumer preferences. For example, in response to the increasing demand for products with better nutritional value and fewer additives, manufacturers may consider reformulating their chips to include healthier oils with better fatty acid profiles or reducing artificial ingredients. By emphasizing the use of natural, high-quality ingredients and clearly labeling products as low in saturated fat or additive-free, manufacturers can appeal to health-conscious consumers who prioritize nutritional content.

Additionally, for companies focused on hygiene and brand quality, maintaining high food safety standards and transparently communicating these standards through marketing channels can significantly increase brand trust. Investing in visible certifications for quality control and hygiene standards and leveraging established brand reputations can be a powerful differentiator in a competitive marketplace. Consumers are more likely to remain loyal to brands they perceive as trustworthy and safe.

Finally, for those targeting price- and affordability-focused consumers, manufacturers can explore strategies to optimize production costs while maintaining acceptable quality. Offering value packs or creating budget-friendly alternatives without compromising nutritional quality can appeal to price-sensitive segments. However, it is also important to balance affordability with perceived quality, ensuring that even low-cost options do not compromise on key attributes such as taste or brand trust.

By aligning product development and marketing strategies with these key consumer priorities, manufacturers can better meet the diverse needs of their target audience and strengthen their market position.

## Figures and Tables

**Figure 1 foods-13-03292-f001:**
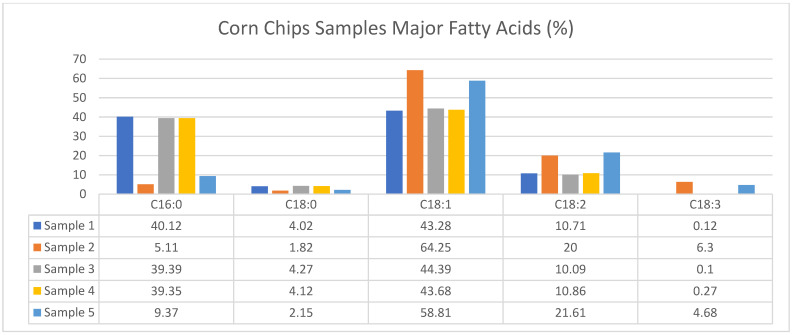
Major fatty acids of corn chips samples.

**Figure 2 foods-13-03292-f002:**
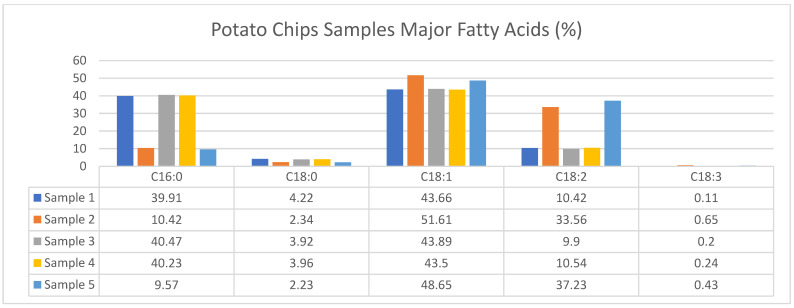
Major fatty acids of potato chips samples.

**Figure 3 foods-13-03292-f003:**
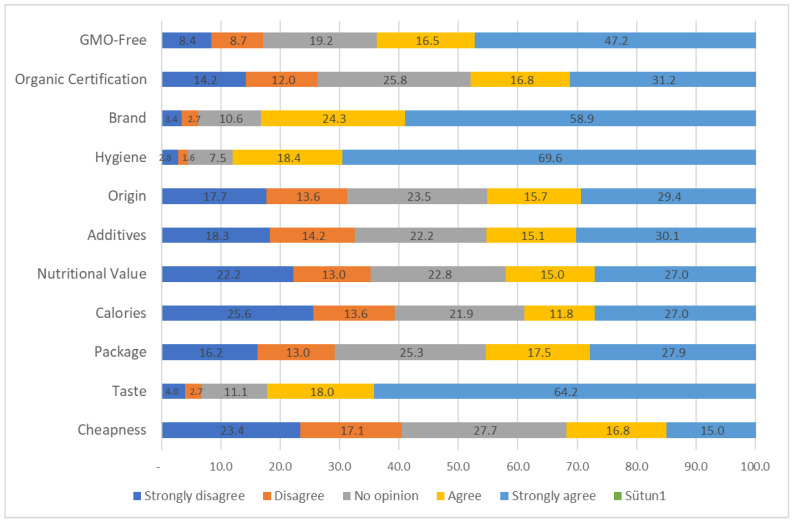
Consumer preferences based on key product attributes (percentage distribution).

**Figure 4 foods-13-03292-f004:**
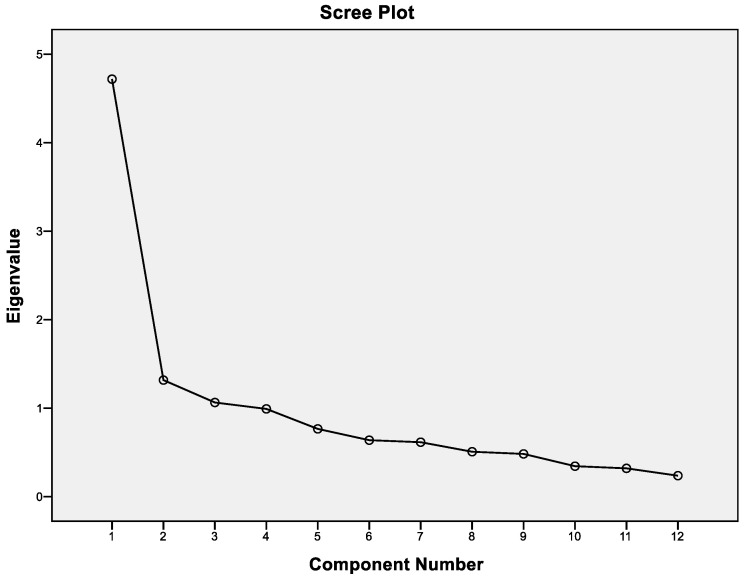
Scree plot.

**Figure 5 foods-13-03292-f005:**
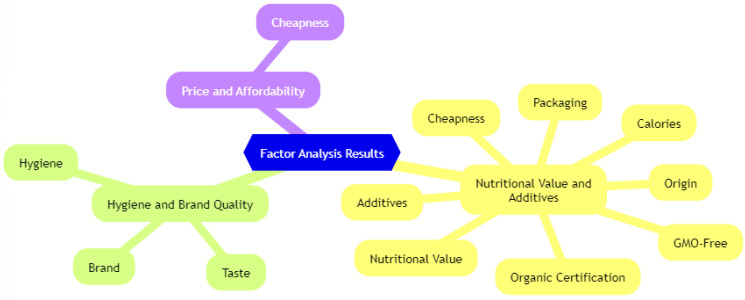
Factors affecting the consumers’ preferences.

**Table 1 foods-13-03292-t001:** Saturated and unsaturated fatty acid profile of corn chips.

Properties	Corn Chips Samples				
	1	2	3	4	5
Fat (%) **	22.09 b	21.31 c	21.86 bc	28.92 a	19.89 d
Protein (%) **	5.25 a	5.26 a	4.95 b	4.93 b	4.90 b
Ash (%) **	3.16 a	0.96 d	2.60 b	3.03 a	1.89 c
Fatty acids (%)					
C12:0 *	0.19 b	Nd	0.26 a	0.14 b	Nd
C14:0 **	0.86 a	Nd	0.86 a	0.86 a	0.15 b
C16:0 **	40.12 a	5.11 d	39.39 b	39.35 b	9.37 c
C18:0 **	4.02 c	1.82 e	4.27 a	4.12 b	2.15 d
C20:0 **	0.35 d	0.55 b	0.37 d	0.50 c	0.91 a
C22:0 NS	Nd	0.27	Nd	Nd	0.46
C24:0	Nd	Nd	Nd	Nd	Nd
C16:1 (*n*-9) *	0.18 b	0.19 b	0.14 b	Nd	0.25 a
C18:1 (*n*-9) **	43.28 e	64.25 a	44.39 c	43.68 d	58.81 b
C18:1t	Nd	Nd	Nd	Nd	Nd
C20:1 (*n*-9) **	0.16 cd	1.50 b	0.12 d	0.22 c	1.61 a
C22:1	Nd	Nd	Nd	Nd	Nd
C18:2 (*n*-6) **	10.71 d	20.00 b	10.09 e	10.86 c	21.61 a
C18:2t	Nd	Nd	Nd	Nd	Nd
C20:2	Nd	Nd	Nd	Nd	Nd
C22:2	Nd	Nd	Nd	Nd	Nd
C18:3 (*n*-3) **	0.12 d	6.30 a	0.10 d	0.27 c	4.68 b
C18:3t	Nd	Nd	Nd	Nd	Nd
TOTAL	99.99	99.99	99.99	100.00	100.00
∑SFA	45.54	7.75	45.15	44.97	13.04
∑MUFA	43.62	65.94	44.65	43.90	60.67
∑PUFA	10.83	26.30	10.19	11.13	26.29
∑UFA	54.45	92.24	54.84	55.03	86.96
∑*n*-3	0.12	6.30	0.10	0.27	4.68
∑*n*-6	10.71	20.00	10.09	10.86	21.61
∑*n*-9	43.62	65.94	44.65	43.90	60.67
PUFA/SFA	0.24	3.39	0.23	0.25	2.02
*n*6/*n*3	89.25	3.17	100.9	40.22	4.62
∑Trans	Nd	Nd	Nd	Nd	Nd

Each value represents the mean of three measurements, and values within the same rows followed by different letters indicate statistically significant differences. NS: not significant, * Significant at *p* < 0.05, ** Significant at *p* < 0.01 according to the LSD (Least Significant Difference) test. Nd: not detected. All values are reported as weight (wt)% of total fatty acid methyl esters. MUFA, monounsaturated fatty acids; SFA, saturated fatty acids; PUFA, polyunsaturated fatty acids; UFA, unsaturated fatty acids.

**Table 2 foods-13-03292-t002:** Saturated and unsaturated fatty acid profile of potato chips.

Properties	Potato Chips Samples				
	1	2	3	4	5
Fat (%) *	34.58 a	28.57 e	29.92 d	31.38 c	32.12 b
Protein (%) *	6.57 b	6.82 a	5.81 d	6.38 b	6.06 c
Ash (%) *	5.57 a	4.35 c	4.47 c	5.37 b	5.22 b
Fatty acids (%)					
C12:0 NS	0.20	Nd	Nd	0.21	Nd
C14:0 *	0.84 b	0.04 c	0.93 a	0.79 b	Nd
C16:0 *	39.91 c	10.42 d	40.47 a	40.23 b	9.57 e
C18:0 *	4.22 a	2.34 c	3.92 b	3.96 b	2.23 d
C20:0 *	0.36 c	0.47 a	0.42 b	0.37 bc	0.39 bc
C22:0 NS	Nd	0.36	Nd	Nd	0.41
C24:0	Nd	Nd	Nd	Nd	0.24
C16:1 (*n*-9) *	0.10 b	0.13 ab	0.17 a	0.16 a	0.12 ab
C18:1 (*n*-9) *	43.66 d	51.61 a	43.89 c	43.50 e	48.65 b
C18:1t	Nd	Nd	Nd	Nd	Nd
C20:1 (*n*-9) *	0.17 c	0.41 b	0.10 d	Nd	0.73 a
C22:1	Nd	Nd	Nd	Nd	Nd
C18:2 (*n*-6) *	10.42 d	33.56 b	9.90 e	10.54 c	37.23 a
C18:2t	Nd	Nd	Nd	Nd	Nd
C20:2	Nd	Nd	Nd	Nd	Nd
C22:2	Nd	Nd	Nd	Nd	Nd
C18:3 (*n*-3) *	0.11 d	0.65 a	0.20 c	0.24 c	0.43 b
C18:3t	Nd	Nd	Nd	Nd	Nd
TOTAL	99.99	99.99	100.00	100.00	100.00
∑SFA	45.53	13.63	45.74	45.56	12.84
∑MUFA	43.93	52.15	44.16	43.66	49.50
∑PUFA	10.53	34.21	10.10	10.78	37.66
∑UFA	54.46	86.36	54.26	54.44	87.16
∑*n*-3	0.11	0.65	0.20	0.24	0.43
∑*n*-6	10.42	33.56	9.90	10.54	37.23
∑*n*-9	43.93	52.15	44.16	43.66	49.50
PUFA/SFA	0.23	2.51	0.22	0.24	2.93
*n*6/*n*3	94.73	51.63	49.5	43.92	86.58
∑Trans	Nd	Nd	Nd	Nd	Nd

Each value represents the mean of three measurements, and values within the same rows followed by different letters indicate statistically significant differences. NS: not significant, * Significant at *p* < 0.01 according to the LSD (Least Significant Difference) test. Nd: not detected. All values are reported as weight (wt)% of total fatty acid methyl esters. MUFA, monounsaturated fatty acids; SFA, saturated fatty acids; PUFA, polyunsaturated fatty acids; UFA, unsaturated fatty acids.

**Table 3 foods-13-03292-t003:** Demographic characteristics.

Variable	Category	Percentage (%)
Age group	18 and under	13.0
	19–24	42.6
	25–34	17.1
	35–50	18.6
	51 and over	8.7
Gender	Male	38.4
	Female	61.6
Education	PhD	1
	Master’s Degree	4.2
	University Graduate	24.6
	University Student	33.6
	High School Graduate	24.1
	Middle School	7.6
	Primary School	4.8
Occupation	Student	43.9
	Private Sector Employee	15.3
	Civil Servant	7.6
	Retired	4.2
	Freelancer	10.5
	Homemaker	3.6
	Teacher	0.6
	Academician	0.9
	Other (Various Professions)	13.3

**Table 4 foods-13-03292-t004:** KMO and Bartlett’s test.

Kaiser–Meyer–Olkin Measure of Sampling Adequacy.		0.879
Bartlett’s Test of Sphericity	Approx. Chi-Square	2893.759
	df	66
	Sig.	0.000

**Table 5 foods-13-03292-t005:** Total variance explained using principal component analysis extraction method.

Component	Initial Eigenvalues			Extraction Sums of Squared Loadings			Rotation Sums of Squared Loadings		
	Total	% of Variance	Cumulative %	Total	% of Variance	Cumulative %	Total	% of Variance	Cumulative %
1	4.718	39.320	39.320	4.718	39.320	39.320	4.058	33.815	33.815
2	1.317	10.976	50.296	1.317	10.976	50.296	1.949	16.244	50.059
3	1.064	8.865	59.161	1.064	8.865	59.161	1.092	9.102	59.161
4	0.991	8.261	67.422						
5	0.765	6.378	73.801						
6	0.638	5.316	79.117						
7	0.615	5.126	84.243						
8	0.507	4.228	88.471						
9	0.483	4.023	92.494						
10	0.344	2.867	95.362						
11	0.320	2.664	98.026						
12	0.237	1.974	100.000						

**Table 6 foods-13-03292-t006:** Rotated component matrix.

	Component		
	1	2	3
Cheapness			0.807
Taste		0.634	0.546
Packaging	0.643		
Calories	0.787		
Nutritional Value	0.850		
Additives	0.829		
Origin	0.730		
Hygiene		0.691	
Brand		0.689	
Organic Certification	0.744		
GMO Free	0.587	0.422	

The extraction was performed using Principal Component Analysis, with the rotation method ap-plied being Varimax and Kaiser Normalization. The rotation process reached convergence in six iterations.

## Data Availability

The consumer survey data have been kept confidential in compliance with personal data protection laws.

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
