# Peer review of "The Effect of Fatty Acids Profile in Potato and Corn Chips on Consumer Preferences"

_foods, 2024, doi:10.3390/foods13203292_

Round 1
Reviewer 1 Report
Comments and Suggestions for Authors
This study aimed to assess the nutritional quality and safety of potato and corn-based chips sold in the local Turkish market. However, there are some weaknesses that need to be addressed.
1. Line 333-335: the manuscript contained none information about the preparation of potato chips and corn chips in the Materials and Methods. Why do you discuss sunflower oil in the Conclusion?
2. Suggest to provide more information about the survey analysis in Materials and Methods section, for example, how do you screen the participants? the information about the participants.
3. Line 218-238: How do you get the data of 3.2 Survey data analysis? Suggest to provide the specific results and data.
4. Line 95: suggest to change “oil content” to “fat content”, most of the discussions in the manuscript use “fat content”.
Author Response
We would like to thank you for your valuable feedback on our manuscript. We appreciate the time and effort you invested in reviewing our work and for your constructive suggestions, which have helped us improve the quality of the manuscript.
Below, we have provided point-by-point responses to each of your comments and highlighted the changes made in the revised version of the manuscript.
Comment 1.
Line 333-335: the manuscript contained none information about the preparation of potato chips and corn chips in the Materials and Methods. Why do you discuss sunflower oil in the Conclusion?
Answer 1.
The mentioned paragraph is corrected according to your suggestions.
Comment 2.
Suggest to provide more information about the survey analysis in Materials and Methods section, for example, how do you screen the participants? the information about the participants.
Answer 2.
We added more information in Materials and Methods section according to your suggestions.
Comment 3.
Line 218-238: How do you get the data of 3.2 Survey data analysis? Suggest to provide the specific results and data.
Answer 3.
We added information under "3.2 Survey data analysis"
Comment 4.
Line 95: suggest to change “oil content” to “fat content”, most of the discussions in the manuscript use “fat content”.
Answer 4.
We have corrected “oil content” to “fat content” in the manuscript.
Reviewer 2 Report
Comments and Suggestions for Authors
The work entitled The Effect of Fatty Acids Profile in Potato and Corn Chips on Consumer Preferences presents interesting results; however, the comparison with the reported literature is not observed; the authors are invited to discuss their results. Also, please see the attached document with the observations.

Author Response
Thank you for your thorough review of our manuscript and for your valuable feedback. We appreciate the time and effort you have taken to provide insightful comments and suggestions, which have been very helpful in improving the quality of our work. Below, we have provided responses to each of your comments and have made the necessary revisions in the manuscript accordingly.
Comment 1
Line 28: Add references and dates about substantial growth
Answer 1
We added dates about substantial growth from the reference of the statement as:
"Since 2020, the snack food industry has experienced substantial growth, propelled by modern, fast-paced lifestyles that make convenience foods more appealing. Chips have emerged as one of the most preferred snacks worldwide [1]."
Reference [1]: Benkhoud, H.; Mrabet, Y.; Nasraoui, N.; Bellazreg, W.; Daly, F.; Chaabane, N.; Hosni, K. Chemical compositions, fatty acid profiles and selected contaminants in commercial potato and corn chips sold in the Tunisian market. Discover Food 2022, 2, 1. https://doi.org/10.1007/s44187-022-00030-8.
Comment 2
Line 36-40: Please add references to support the ideas
Answer 2
We added new reference to support the ideas.
Reference [10]: Santeramo FG, Carlucci D, De Devitiis B, Seccia A, Stasi A, Viscecchia R, Nardone G. Emerging trends in European food, diets and food industry. Food Res Int. 2018;104:39-47. DOI: 10.1016/j.foodres.2017.10.039
Comment 3
Line 73: Please explain the number 667 and how it was obtained; the authors are sure that the number is enough for an entire country
Answer 3
We explained the number 667 according to your suggestion in Equation 1.
The sample size 667 is enough for analysis in this paper. Also Kaiser-Meyer-Olkin (KMO) measure result of 0.879 confirms that the sample size is sufficient for analysis as indicated in Table 3[25].
Reference [25] Hair, J.F.; Anderson, R.E.; Tatham, R.L.; Black, W.C. Multivariate Data Analysis, 5th ed.; Pearson Education, Inc.: New Delhi, India, 2003.
Comment 4
Line 171: Please improve the figure1 and 2
Answer 4
We improved the figures 1 and 2 according to your suggestion by adding datatable under the bars.
Comment 5
The work presents interesting results; however, in the results section, the comparison with the reported literature is strange; the authors are invited to discuss their results, and probably, the greatest number of reports will not be found in the field of snacks. However, other foods exist and will allow us to have a general point of view about fatty acids.
Answer 5
In line with your recommendations, we have revised the results section by expanding the discussion and supporting it with relevant sources.
We added these paragraphs:
"In our research, none of the potato and chip samples examined contained trans fatty acids. However, in [12], the TFA concentrations varied across the samples, with mean lev-els ranging from 1.41 g/100 g of fat to 2.88 g/100 g of fat and 0.41 g/100 g of food to 0.78 g/100 g of food.
Our analysis revealed that the fatty acid profile of potato chips varied significantly depending on the frying medium, with sunflower oil leading to a higher absorption of saturated fatty acids compared to oleogels. This is consistent with findings from studies using rice bran and beeswax oleogels, which showed a reduction in oil uptake and im-proved frying stability[14,29]. Additionally, the formation of starch-lipid complexes in chips fried in sunflower oil was less pronounced than in those fried in oleogels, similar to the results of studies on intermittent frying and the impact of unsaturated oils [30]. Fur-ther, our data showed that chips fried in oleogels absorbed 23% less oil compared to those fried in conventional oils, consistent with prior research [14]. The use of rice bran wax as an oleogelator proved effective in reducing fat uptake and maintaining sensory quality during frying, as previously demonstrated [31]."
Comment 6
Also, I suggest that the author include the study's limitations.
Answer 6
We added the following paragraph to the results section.
"The analysis is limited to a sample of commercially available chips in Turkey, which may not fully represent global market variations. Additionally, the study focuses primari-ly on potato and corn chips, potentially excluding other snack categories that could offer further insights into fatty acid compositions. "
Reviewer 3 Report
Comments and Suggestions for Authors
I have reviewed the perspective manuscript titled: The Effect of Fatty Acids Profile in Potato and Corn Chips on Consumer Preferences.
This article aims to identify and compare fatty acid profiles of commercially available five corn and potato brand chips are analyzed and to evaluate their impacts on consumer preferences in Turkey by 667 consumers’ purchase preferences about the nutritional quality. The information of this work is useful and relevant and there is many valuable guidance for manufacturers and consumers developing snack options for healthier eating habits and enhancing the health-promoting properties of the manuscript that could be adapted by food processing industry especially for snack industries in the future. I think the manuscript is acceptable after minor revision as attached revised suggestion. Although, all the cited journals were not in abbreviated format, it still contains interesting and useful information for consumers and snack industries. Abstract is well written upon and the results for notable differences in nutritional content between corn and potato chips and their survey results are given. Introduction is well addressed including chips market, the risks of various health problems, consumer attitudes and preferences in the world and Turkey. The information of the fatty acid composition of corn and potato chips and their impacts on consumer preferences were introduced well in this manuscript.
I am not a native English speaker. The manuscript seems do not have major mistakes are detected except all journal names are not abbreviated in reference section, although most of them are presented following the format of Foods. However, the perspective manuscript can be understood. I suggested the information at page 6 line 184 “more minerals” should add more information. Is high mineral content implying more sodium chloride? Cited references are well discussed.
I enjoyed reading this manuscript. This manuscript also presents some nutrition information for their scaling-up feasibility.
Is the manuscript scientific? Yes, the manuscript is scientific.
-Is the introduction of the manuscript well-organized and adequate?
And the introduction section is adequate and well-organized for snacks but it does including the information of high mineral content snacks.
-Is the method part of the manuscript appropriate?
The analyzed method of this manuscript is appropriate!
-What does it add to the subject area compared with other published material?
The snack foods are not much papers publishing further investigate the fatty acid composition of corn and potato chips and their impact on consumer preferences particularly in different markets. Therefore, it is a new area compared to other published materials.
-Are the conclusions consistent with the evidence and arguments presented? Yes, the conclusions are consistent with the arguments and evidence and well address the developing better snack options for processing snacks except the information of high mineral content chips.
-Do they address the main question posed?
Yes, the authors address the main questions.
Author Response
Thank you for your thorough review of our manuscript and for your valuable feedback. We appreciate the time and effort you have taken to provide insightful comments and suggestions, which have been very helpful in improving the quality of our work. Below, we have provided responses to each of your comments and have made the necessary revisions in the manuscript accordingly.
Comment 1
The journal names in the reference section are not abbreviated.
Answer 1
We have reviewed and updated the reference section to ensure that all journal names are now properly abbreviated, in accordance with the journal's formatting guidelines.
Comment 2
The phrase "more minerals" on page 6, line 184 should be expanded upon. Is the high mineral content referring to more sodium chloride?
Answer 2:
We have revised the text to provide more specific information about the minerals present, clarifying that it refers to a range of minerals including potassium, calcium, and magnesium.
Our revised text emphasizes that potato chips generally contain more minerals compared to corn chips, highlighting the presence of potassium, calcium, phosphorus, and magnesium.
Also we discussed the potential health benefits of these minerals, including their role in maintaining overall health, despite the high fat content of the chips.
Round 2
Reviewer 2 Report
Comments and Suggestions for Authors
Thank you for your comments and corrections
Author Response
Dear Reviewer,
We would like to express our gratitude for your review of our. We appreciate your positive feedback and the time you took to evaluate our work.
We noticed that all criteria were rated as "Yes," and there were no additional comments suggesting changes or further improvements. We would be happy to provide any further clarifications or revisions if needed, as we aim to ensure our manuscript meets all the expectations for publication.
Thank you again for your thoughtful review.